# Performance of a Novel Real-Time PCR-Based Assay for Rapid Monkeypox Virus Detection in Human Samples

**DOI:** 10.3390/microorganisms11102513

**Published:** 2023-10-08

**Authors:** Flora Marzia Liotti, Simona Marchetti, Federico Falletta, Sara D’Onghia, Maurizio Sanguinetti, Rosaria Santangelo, Brunella Posteraro

**Affiliations:** 1Dipartimento di Scienze di Laboratorio e Infettivologiche, Fondazione Policlinico Universitario A. Gemelli IRCCS, 00168 Rome, Italy; floramarzia.liotti@policlinicogemelli.it (F.M.L.); simona.marchetti@policlinicogemelli.it (S.M.); sara.donghia@policlinicogemelli.it (S.D.); rosaria.santangelo@unicatt.it (R.S.); 2Dipartimento di Scienze Biotecnologiche di Base, Cliniche Intensivologiche e Perioperatorie, Università Cattolica del Sacro Cuore, 00168 Rome, Italy; federico.falletta@analisibio.it (F.F.); brunella.posteraro@unicatt.it (B.P.); 3Dipartimento di Scienze Mediche e Chirurgiche Addominali ed Endocrino Metaboliche, Fondazione Policlinico Universitario A. Gemelli IRCCS, 00168 Rome, Italy

**Keywords:** monkeypox virus, PCR assay, rapid detection, human samples

## Abstract

The ongoing epidemic of mpox, namely human monkeypox virus (MPXV) infection, requires rapid and reliable laboratory diagnosis. We report on the QIAstat-Dx viral vesicular panel PCR assay that allows the detection of (within 75 min) six vesicular disease-causing viruses, including MPXV. We analyzed 168 clinical samples, known to be positive (51 samples) or negative (117 samples) for MPXV clade II, obtained from patients at their mpox diagnosis or follow-up. QIAstat assay results were compared to those of a MPXV-specific reference PCR assay. The QIAstat assay detected MPXV (clade II) in 51 (100%) of 51 samples and did not detect MPXV in 117 (100%) of 117 samples, resulting in a positive or negative agreement of 100% (95% CI, 93.0–100) and 100% (95% CI, 96.8–100), respectively. Of the 20 patients diagnosed with mpox, 18 (90.0%) had at least a vesicular swab and 1 (5.0%) had only an oropharyngeal swab positive for MPXV. At mpox follow-ups, 2 (10.0%) of 20 patients had first-time positive whole blood samples. Thirteen MPXV-negative samples were positive for mpox-mimicking viruses. Our findings show the excellent performance of the QIAstat-Dx assay for MPXV detection in clinical samples. Further studies are needed before considering a large-scale application of the QIAstat-Dx assay.

## 1. Introduction

As of July 2022, the World Health Organization (WHO) declared the worldwide epidemic of mpox (formerly called monkeypox, i.e., the human monkeypox virus [MPXV] infection), as a global public health emergency [1]. Consistent with this, the European Centre for Disease Prevention and Control (ECDC) and the WHO Regional Office for Europe reported 25,893 mpox cases in Europe between March 2022 and August 2023 (https://monkeypoxreport.ecdc.europa.eu/; accessed on 11 September 2023). Of these cases, 25,714 were laboratory-confirmed and 487 cases, where sequencing was available, were identified as MPXV clade II (formerly known as the West African clade), which is distinguished from MPXV clade I (formerly known as the Central African clade) [2]. As in other previous zoonotic viral epidemics/pandemics [3], mpox emergence has prompted commercial or academic developers of diagnostic assays to expand the capacity for MPXV detection in public health or clinical microbiology laboratories [4,5].

The diagnosis of mpox based on clinical presentation alone may be difficult, especially in atypical presentation cases [6]. Therefore, it is important not only to differentiate the two MPXV clades from each other, but also MPXV from other orthopoxviruses (MPXV is a double-stranded DNA virus belonging to the genus Orthopoxvirus [OPX]) [7,8,9] or other vesicular disease-causing viruses such as herpes simplex virus 1 and 2 (HSV-1/HSV-2), human herpesvirus 6 (HHV6), enterovirus (EV), or varicella zoster virus (VZV) [10,11]. The speed of diagnosis, achievable through a point-of-care testing (POCT) molecular assay, may be important, especially if the surge in mpox cases requires easier detection and timely control at the local level to decrease the global spread of the disease [4].

To date, the QIAstat-Dx viral vesicular panel assay (Qiagen, Germantown, MD, USA), which launched on the market in September 2022 (https://www.rapidmicrobiology.com/news/monkeypox-health-emergency-qiagen-launches-syndromic-test-panel; accessed on 11 September 2023) is the only commercially available real-time PCR assay to be both a POCT (approximately 1 h of reaction time required) and a multitarget panel (MPXV and five other vesicular disease-causing viruses included) assay. It should be noted that the assay (hereafter referred to as QIAstat-Dx assay) is a research use only (RUO) assay whose clinical utility for the detection of MPXV (and other (co)-infecting viruses) has been shown in two evaluation studies published so far [10,11]. In one of these studies [11], human samples other than skin lesion material, which remains the preferred clinical sample based on the WHO’s mpox laboratory testing guidelines [12], were used. Accordingly, testing multiple samples, including those from the oropharyngeal tract (which the WHO recommends as an additional diagnostic sample [12]) or blood (which the WHO considers as an investigational sample [12]), may be helpful for mpox diagnosis [13,14,15,16].

Here, we report the performance evaluation of the QIAstat assay using a collection of clinical (vesicular swab, oropharyngeal swab, and blood) samples, obtained from July 2022 to August 2023 and previously classified as positive or negative for MPXV. Results from the QIAstat assay were compared to those of a reference real-time PCR assay that uses a MPXV-specific primers/probe combination previously described by the Centers for Disease Control and Prevention (CDC) [8,17] and were used to calculate inter-assay agreement.

## 2. Materials and Methods

### 2.1. Study Setting and Clinical Samples

This retrospective study included 168 clinical (63 whole blood, 58 oropharyngeal swab, and 47 vesicular swab) samples collected between July 2022 and August 2023 at the Fondazione Policlinico Universitario A. Gemelli IRCCS, a large hospital in Rome (Italy). Patients from whom samples were obtained had a suspicion (40 patients) or had a follow-up (20/40 patients) for mpox. All but 2 patients (median (interquartile range) age, 40 (31–46) years) were male, and 19 patients had underlying HIV (8/40 [20.0%] patients), *Treponema pallidum* (5/40 [12.5%] patients), HIV and *T. pallidum* (3/40 [7.5%] patients), or *Leishmania* (3/40 [7.5%] patients) infections. Samples were partly from patients at the time of mpox diagnosis (117 samples) and partly from patients quarantined for a median (interquartile range) time of 10 (7–14) days since mpox diagnosis (51 samples). Part of this study has been presented at the 33rd European Congress of Clinical Microbiology and Infectious Diseases (ECCMID) held in Copenhagen, Denmark (15–18 April 2023).

Since May 2022, when the first mpox case was identified in Italy, the clinical microbiology laboratory of the above-mentioned hospital has joined the national mpox surveillance network of the Italian Ministry of Health (https://www.trovanorme.salute.gov.it/norme/renderNormsanPdf?anno=2022&codLeg=88439&parte=1%20&serie=null; accessed on 11 September 2023). Accordingly, we have developed and validated an in-house mpox diagnostic algorithm based on the RealStar Orthopoxvirus PCR kit 1.0 (Altona Diagnostics, Hamburg, Germany) and Quanty Monkeypox (Clonit S.r.l, Milan, Italy) real-time PCR assays. In particular, the RealStar assay detects zoonotic OPX species including MPXV, whereas the Quanty assay specifically detects MPXV. As shown in Appendix A, positive samples at the screening with the first assay underwent confirmatory testing with the second assay, which can discriminate between MPXV clade I and clade II.

Of 168 samples studied, 51 were positive for MPXV (clade II) DNA and 117 were negative for MPXV DNA according to the in-house mpox diagnostic algorithm results. All swab samples originally collected into universal transport medium (UTM; Copan, Brescia, Italy) or whole blood samples originally collected in K_2_-EDTA tubes (BD Vacutainer; Becton Dickinson, Rome, Italy) were portioned in aliquots that were kept frozen at −80 °C until testing with both reference and QIAstat assays (see below).

### 2.2. Reference Assay Testing

A method developed by Li et al. [17] that targets the MPXV G2R_G-encoding gene was used as a comparator. Briefly, DNA was extracted from a 400 µL sample’s aliquot using the EZ1 Advanced XL (Qiagen) automated platform with the Qiagen EZ1 DSP virus kit, according to the manufacturer’s instructions. Part of the resulting 120 μL DNA solution (5 μL) was added to the real-time PCR reaction mixture, which contained the following oligonucleotide primer pair and probe: MPXV-forward primer 5′-GGAAAATGTAAAGACAACGAATACAG, MPXV-reverse primer 5′-GCTATCACATAATCTGGAAGCGTA, and MPXV-probe 5′FAM-AAGCCGTAATCTATGTTGTCTATCGTGTCC-3′BHQ1. PCR was performed using a CFX96 Touch Real-Time PCR Detection System (Bio-Rad, Hercules, CA, USA) instrument and the following cycling conditions: 1 cycle at 95 °C for 6 min and 45 cycles at 95 °C for 5 s and 60 °C for 20 s. A positive result (i.e., a cycle threshold [Ct] less than 40) for the viral target indicates the presence of MPXV DNA in the patient sample.

### 2.3. QIAstat-Dx Assay Testing

A 300 µL sample’s aliquot was subjected to the QIAstat-Dx assay, which has been ideated to detect viral nucleic acid from transport media (i.e., UTM) by real-time reverse-transcription PCR. Briefly, the assay consists of a single-use multiplex PCR syndromic cartridge that includes all reagents needed for the nucleic acid extraction, nucleic acid amplification, and detection of 7 targets (MPXV clade I, MPXV clade II, HSV-1, HSV-2, HHV6, EV, and VZV). The assay contains an internal RNA control that serves to confirm successful completion of all steps of the detection process. For each sample, the cartridge was loaded onto the QIAstat-Dx Analyzer 1.0 (Qiagen) instrument and results were available in 75 min. A positive result (i.e., a Ct less than 40) for at least one of viral targets indicates the presence of MPXV and/or other virus(es) in the patient sample.

### 2.4. Data Analysis

We calculated positive percent agreement (PPA) and negative percent agreement (NPA), together with their respective confidence intervals (CIs), for the QIAstat-Dx assay results in comparison with the reference assay results. Statistical analysis was performed with the SPSS statistics software version 24.0 (IBM Corp., Armonk, NY, USA). Differences between the Ct values (expressed as mean ± standard deviation [SD]) in sample groups were assessed using the Student’s *t*-test or a one-way analysis of variance (ANOVA) with the Tukey’s multiple-comparison test, as appropriate. Two-sided *p* values of <0.05 were considered statistically significant.

## 3. Results

As shown in Table 1, using the QIAstat-Dx assay, 51 (30.4%) of 168 clinical samples tested in total had a positive result for MPXV (clade II). Of these, 23 (45.1%) were vesicular swab samples, 17 (33.3%) were oropharyngeal swab samples, and 11 (21.6%) were whole blood samples. The 51 samples were from 20 patients who had received a laboratory-confirmed mpox diagnosis upon first admission (39 samples) or at the follow-up (12 samples) at our hospital. Five samples were also positive for HSV-2 (1 sample), HHV6 (2 samples), or EV (2 samples). Additionally, 13 samples were negative for MPXV but positive for HSV-1 (2 samples), HHV6 (4 samples), EV (2 samples), or VZV (5 samples). Except for MPXV-positive samples (see below), no testing was performed in this study to confirm positive QIAstat-Dx assay results for vesicular disease-causing viruses other than MPXV.

The QIAstat-Dx assay results of the 20 positive patients for MPXV were analyzed based on the time of sample collection. As shown in Appendix A, mpox diagnosis was based on a positive vesicular swab in 18 (90.0%) patients and on a positive oropharyngeal swab in 2 (10.0%) patients. MPXV was detected in the whole blood samples from 8 (40.0%) of 20 patients; of these, 6 patients also had MPXV detected in both vesicular swab and oropharyngeal swab samples and the remaining patients in a vesicular (1 patient) or oropharyngeal (1 patient) swab sample.

Regarding patients diagnosed with mpox (n = 20), the number of samples collected in total was 108, including 57 samples at diagnosis, 38 samples at the first follow-up (i.e., at a median (IQR) time of 8 (7–12) days after diagnosis), and 13 samples at the second follow-up (i.e., at a median (IQR) time of 17 (12–20) days after diagnosis). Regarding patients not diagnosed with mpox (n = 20), the total number of samples collected was 60, all of which were samples used to exclude mpox.

We compared the QIAstat-Dx assay results with those of the reference assay for MPXV detection. As shown in Table 2, QIAstat-Dx assay detected MPXV in 51 (100%) of 51 MPXV-positive samples by the reference assay and did not detect MPXV in 117 (100%) of 117 MPXV-negative samples by the reference assay. The agreement between the assays was 100% (95% CI, 93.0–100) for positive samples and 100% (95% CI, 96.8–100) for negative samples.

Considering all 51 MPXV-positive results (23 for vesicular swab samples, 17 for oropharyngeal swab samples, and 11 for whole blood samples; see also Appendix A for details), the mean (±SD) Ct values were 29.7 ± 6.4 for the QIAstat-Dx assay and 28.4 ± 6.1 for the reference assay (*p* = 0.59). Stratifying QIAstat-Dx assay’s Ct values by type of samples (Figure 1) revealed that the mean (±SD) Ct value for vesicular swab samples (27.3 ± 6.9) differed significantly from the mean (±SD) Ct value for oropharyngeal swab samples (29.1 ± 5.4) and from the mean (±SD) Ct value for whole blood samples (35.5 ± 2.7) (*p* < 0.001). Likewise, stratifying the reference assay’s Ct values by type of samples (Figure 1) revealed that the mean (±SD) Ct value for vesicular swab samples (25.7 ± 5.9) differed significantly from the mean (±SD) Ct value for oropharyngeal swab samples (27.9 ± 5.2) and from the mean (±SD) Ct value for whole blood samples (34.7 ± 2.7) (*p* < 0.001).

Of the 51 MPXV-positive samples, 39 (18 vesicular swab samples, 13 oropharyngeal swab samples, and 8 whole blood samples) were found at mpox diagnosis and 12 (5 vesicular swab samples, 4 oropharyngeal swab samples, and 3 whole blood samples) were at an mpox follow-up (Appendix A). In the first sample group (n = 39), the mean (± SD) Ct values were 28.4 ± 5.8 for the QIAstat-Dx assay and 27.4 ± 5.5 for the reference assay (*p* = 0.59). In the second sample group (n = 12), the mean (± SD) Ct values were 33.1 ± 6.8 for the QIAstat-Dx assay and 31.7 ± 6.9 for the reference assay (*p* = 0.89). For both assays, no statistically significant differences were also observed when the Ct values of the samples at mpox diagnosis were compared to the Ct values of the samples at an mpox follow-up (QIAstat-Dx assay, 28.4 ± 5.8 versus 33.1 ± 6.8 [*p* = 0.63]; 27.4 ± 5.5 versus 31.7 ± 6.9 [*p* = 0.27]).

## 4. Discussion

In this retrospective evaluation of the QIAstat-Dx assay for the detection of MPXV in human sample types, the assay showed 100% agreement with the reference assay for not only 51 MPXV (clade II) positive samples but also for 117 MPXV (clade I/clade II) negative samples. The QIAstat-Dx assay results were also in full agreement with those obtained using a two-assay mpox diagnostic algorithm (currently in use in our clinical microbiology) for all 168 samples tested. Although MPXV-positive vesicular swabs allowed for the diagnosis of mpox in 18 (90.0%) of 20 patients, the oropharyngeal swab was the only MPXV-positive sample in 1 (5.0%) of 20 patients. When only considering the samples collected at an mpox follow-up, whole blood samples were positive for the first time in 2 (10.0%) of 20 patients.

To our knowledge, this is the third published study to report results on the QIAstat-Dx assay. In the study by Wilber et al. [10], QIAstat-Dx allowed researchers to detect MPXV clade II in 36 of 47 samples, whereas the comparator (Mpox Virus DNA Qualitative Real-Time PCR assay; Quest Diagnostics, Chantilly, VA, USA) detected MPXV clade II in 37 of 47 samples, thus resulting in a positive agreement of 97.3% (36/37 samples). The sample with a discordant result had a Ct value with the Quest Diagnostics assay near the limit of detection, suggesting a sample’s viral load below the limit of detection of the QIAstat-Dx assay. As in our study, the agreement was 100% for negative (11/11) samples. In the study by Batty et al. [11], QIAstat-Dx allowed for the detection of MPXV clade II in 28 of 124 samples (previously known to be positive for at least one of seven viral targets included in the assay), thus resulting in a positive agreement of 100% (28/28 samples) with a laboratory-developed MPXV-specific reference assay. Using the 124 samples as respective negative controls for each viral target, the agreement was 100% for negative (96/96) samples.

Because we focused on the PCR-based diagnosis of mpox, unlike Batty et al. [11], we did not develop specific reference assays for viral targets other than MPXV in our study. However, excluding 13 (72.2%) of 18 samples in which MPXV was co-detected, the QIAstat-Dx assay would have allowed for the diagnosis of non-MPXV (HSV-2, HHV6, or EV) infections when patients were sampled for mpox suspicion. Notably, Batty et al. [11] reported a positive agreement of 89.0% (16/18 samples) for HSV-1 and 100% (16/16 samples) for VZV, whereas 18 (82.0%) of 22 HHV6-positive samples had a positive result with the QIAstat-Dx assay. It should be noted that all 22 samples were plasma samples [11], supporting the hypothesis that using a non-validated sample type such as plasma might result in (false)-negative results with the QIAstat-Dx assay. Batty et al. [11] were, unfortunately, unable to test HHV6-positive samples other than plasma samples, although they used 98 (positive for other viral targets than HHV6) samples from multiple anatomical sites (including skin and oropharynx). Another explanation of (false) negativity could be the (expected) relatively lower viral load in samples other than lesions (i.e., vesicles, pustules, and crusts, eventually present in the genital, anal, or oral sites) [12].

To counterbalance the risk of failing to diagnose mpox, clinicians have been increasingly encouraged to collect more than one sample [13,14,15,16], possibly consisting of the triad of lesion, oropharyngeal swab, and blood samples. With the aim to elucidate the diagnostic role of multisite sampling, Rizzo et al. [15] analyzed 966 samples (including 252 lesion samples and 278 oropharyngeal swabs among others) from 625 patients with clinical evidence of mpox. Similar to this study, but with a larger sample size (we analyzed 168 samples from 40 patients), the authors found that 15 (5.2%) of 286 patients with MPXV-positive samples had a negative lesion sample (though in the presence of vesicles/pustules/crusts); of 15 patients, 11 had MPXV detected in oropharyngeal swab samples (positive alone in 6 patients or positive together with other sample types in 5 patients) and 4 had MPXV only detected in anal samples (2 in the presence of lesions) [15]. While showing the lesion material as the sample yielding the highest positivity rate for MPXV (93.2% positive of samples tested), which was substantially higher than that observed in our study (48.9% positive of samples tested), according to previous evidence [18], Rizzo et al. [15] suggested that oropharynx and anus should be sampled, in combination with the material from lesions, in patients with high suspicion for mpox.

We noted differences between the Rizzo et al.’s study [15] and our study regarding the clinical/epidemiological context (samples collected from May 2022 to November 2022 instead of from July 2022 to August 2023) and the PCR-based assays, respectively, used for MPXV detection. However, both studies agree that lesion swabs (vesicular swabs in our study) had lower Ct values, which reflect the presence of higher viral loads [13], in PCR-based assays. Not surprisingly, plasma samples (whole blood samples in our study) showed the highest Ct values (median (IQR), 34 [(31–35) in the Batty et al.’s study [11] and mean value (SD), 35.5 ± 2.7 in our study). Although the clinical significance of viremia is unclear [6], blood samples may hold value in the early days of clinical presentation [16] or for viral load monitoring [13]. Unlike previous studies [10,11], we included whole blood (not plasma) as a non-swab biological matrix for the QIAstat-Dx assay. We know that molecular assays can differ in efficiency depending on real-time PCR chemistry and cycling conditions. Thus, like Batty et al. [11] and unlike Wilber et al. [10], we evaluated the QIAstat-Dx performance not by comparing it to the commercial PCR assays marketed shortly after May 2022 (when the first mpox cases were reported to the WHO [1]) but using a CDC-developed assay [17] as a reference assay. We observed that the Ct values with the QIAstat-Dx and reference assays were comparable and were significantly lower in vesicular swabs compared to other types of samples (including whole blood samples). Additionally, we observed that several samples were still positive at mpox follow-up (when viral loads were presumably lower than those of samples at mpox diagnosis), underscoring the reliability of the QIAstat-Dx assay for MPXV detection. As no consensus is currently available on the preferred sample type for the QIAstat-DX assay, full agreement between the QIAstat-DX assay and reference assay results for all samples in our study should indicate no bias in any type of sample tested with the QIAstat-DX assay.

This study is limited by the relatively small number of samples included and its retrospective nature that involved the analysis of frozen aliquots of samples originally collected from patients. While the absence of discrepancies between QIAstat-Dx and reference assays’ results in our study were caused by the sample aliquots being tested simultaneously, we are confident that the sensitivity of both assays would be the same if the tests were conducted with fresh, rather than frozen, samples from each patient. Secondly, the lack of detailed patient data (presence of symptoms, symptom onset, etc.) prevented us from correlating sample positivity with disease stages and understanding the value of sample types, such as whole blood samples, for the diagnosis of mpox.

In conclusion, we showed the potential of the QIAstat-Dx assay as an easy-to-use, rapid, and high-performance real-time PCR assay for the detection of MPXV in different sample types; however, whether this assay can be considered a POCT diagnostic tool for large-scale applications remains to be established. The QIAstat-Dx offers the possibility of excluding mpox and simultaneously diagnosing non-mpox vesicular infections, which may be important not only in view of the syndromic diversity of mpox but also of common (HSV, VZV, etc.) mpox-mimicking viruses. Finally, our experience with mpox diagnostics (including the QIAstat-Dx assay) reinforces the centrality of the clinical microbiology laboratory to develop new (in-house) assays or refine existing (commercial) assays.

## Figures and Tables

**Figure 1 microorganisms-11-02513-f001:**
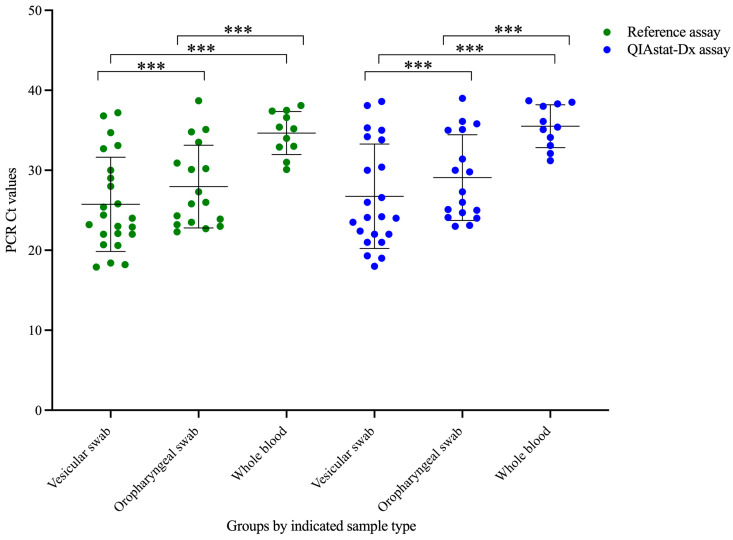
Distribution of Ct values for 51 clinical samples tested with both reference and QIAstat-Dx real-time PCR assays. For each assay, values were stratified by sample type (vesicular swab, oropharyngeal swab, or whole blood). In each scatter dot plot, the central line indicates the mean Ct value and the area between solid lines indicates the standard deviation value. Asterisks indicate statistical significance (*p* < 0.001) between groups as assessed using a one-way analysis of variance (ANOVA) with the Tukey’s multiple-comparison test. Ct—cycle threshold.

**Table 1 microorganisms-11-02513-t001:** Results of 168 clinical samples tested for the 7 viral targets included in the QIAstat-Dx assay.

Type of Samples (No. Tested)	MPXV		HSV-1		HSV-2		HHV6		EV		VZV	
	Pos	Neg	Pos	Neg	Pos	Neg	Pos	Neg	Pos	Neg	Pos	Neg
Vesicular swab (47)	23	24	1	46	0	47	0	47	1 ^1^	46	3	44
Oropharyngeal swab (58)	17	41	1	57	1 ^2^	57	3 ^3^	55	2	56	1	57
Whole blood (63)	11	52	0	63	0	63	3 ^3^	60	1 ^1^	62	1	62
All samples (168)	51 ^4^	117 ^5^	2	166	1	167	6	162	4	164	5	163

^1^ Co-detected with MPXV. The two MPXV/EV-positive samples were from two different patients. ^2^ Co-detected with MPXV. ^3^ Co-detected with MPXV in 1 of 3 samples, respectively. The two MPXV/HHV6-positive samples were from two different patients. ^4^ Includes 5 samples that were also positive for viruses other than MPXV (2 for EV, 1 for HSV-2, and 2 for HHV6; see footnotes 1 to 3). ^5^ Includes 13 samples negative for MPXV but positive for viruses other than MPXV.

**Table 2 microorganisms-11-02513-t002:** Performance of the QIAstat-Dx assay for detection of MPXV in clinical samples.

QIAstat-DX Assay	Reference Assay		
	Positive Results	Negative Results	Total Results
Positive results	51	0	51
Negative results	0	117	117
Total results	51	117	168
Positive % agreement	100 (95% CI, 93.0–100)		
Negative % agreement	100 (95% CI, 96.8–100)		

## Data Availability

Data may be available upon reasonable request.

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
