# Peer review of "Performance of a Novel Real-Time PCR-Based Assay for Rapid Monkeypox Virus Detection in Human Samples"

_microorganisms, 2023, doi:10.3390/microorganisms11102513_

Round 1

Reviewer 1 Report

Monkey Pox Viral Load Assay Review

The authors of this paper address the utility and need for viral load assays as a crucial tool for rapid and reliable laboratory diagnostics in response to the ongoing human monkeypox virus (MPXV) epidemic. They conducted a retrospective study using 168 clinical samples from MPXV clades I and II to evaluate the QIAstat-Dx Viral Vesicular panel PCR assay (QIAstat-Dx Assay) as a potential tool to fulfill this need and compared it to a reference MPXV-specific PCR assay. Their findings revealed that the QIAstat-Dx assay exhibits 100% accuracy in distinguishing known positives from known negatives for MPXV. The assay's performance is comparable to the reference MPXV-specific PCR assay. Additionally, the QIAstat-Dx assay can rapidly detect six viral vesicular viruses, including MPXV, demonstrating its versatility as a viral load assay. The authors propose their findings and reference results from others to demonstrate the assay's effectiveness on clinical samples and showcase the potential of the QIAstat-Dx assay to expand beyond its research-only application (RUO) to large-scale applications within clinical settings.

This paper provides valuable insights into the significance of having viral load assays capable of detecting specific targets as a tool for diagnosis, especially in epidemic scenarios. The authors advocate for the QIAstat-Dx assay as a rapid and reliable solution for detecting MPXV in infected individuals across various sample types. They utilize the significant difference in MPXV CT values among the sample types to reinforce the importance of collecting multiple sample types for MPXV diagnosis, recognizing variations in detection across different tissue types. While their approach involves comparing clinical MPXV samples among different clades and between positive and negative cases, showcasing the robustness of the assay, the addition of non-clinical positive or negative controls alongside their internal RNA control could have further strengthened their findings. The paper maintains a well-organized structure and consistent flow in its text and figures. Overall, the data presented in this paper is suitable for the conclusions presented by the authors. Though the paper is well thought out and thorough, I outline some concerns in the section below:

Minor Changes/Questions to consider:

1.     In lines 195-197, the text describes a significant difference in the mean of the CT values among various sample types, with the results presented in Figure 1. However, Figure 1 displays a significance bar above VS and WB, implying a significant difference between these sample types. To address this, I recommend adding additional significance bars to illustrate any significant differences between VS and OS and between OS and WB to highlight variations in mean CT values among different sample types. This would help Figure 1 more accurately align with what is stated in the text.

2.     The acronyms for vesicular swabs (VS), oropharyngeal swabs (OS), and whole blood (WB) are introduced relatively late in the paper. They initially appear in the description for Figure 1 (Lines 208 and 209), even though they are used earlier in Table 2 (Line 173) without prior introduction in the text. This can potentially lead to confusion among readers trying to deduce their meaning. To address this issue, I recommend either introducing the acronyms earlier in the text, possibly in the first few lines of the Materials and Methods section, or eliminating the acronyms for each sample type in Table 2 (e.g., replace "VS" with "Vesicular swab"). These adjustments would enhance clarity throughout the paper.

3.     Remove parenthesis for “MPXV (and other…” in line 132, as it is unnecessary since there is no closing parenthesis for the statement.

-         

Author Response

  1. In lines 195-197, the text describes a significant difference in the mean of the CT values among various sample types, with the results presented in Figure 1. However, Figure 1 displays a significance bar above VS and WB, implying a significant difference between these sample types. To address this, I recommend adding additional significance bars to illustrate any significant differences between VS and OS and between OS and WB to highlight variations in mean CT values among different sample types. This would help Figure 1 more accurately align with what is stated in the text.

Answer: We improved Figure 1 by adding additional significance bars to highlight the variations in mean CT values between VS and OS and between OS and WB. This makes Figure 1 more accurately aligned with the relative statement in the text. See revised Figure 1.

  1. The acronyms for vesicular swabs (VS), oropharyngeal swabs (OS), and whole blood (WB) are introduced relatively late in the paper. They initially appear in the description for Figure 1 (Lines 208 and 209), even though they are used earlier in Table 2 (Line 173) without prior introduction in the text. This can potentially lead to confusion among readers trying to deduce their meaning. To address this issue, I recommend either introducing the acronyms earlier in the text, possibly in the first few lines of the Materials and Methods section or eliminating the acronyms for each sample type in Table 2 (e.g., replace "VS" with "Vesicular swab"). These adjustments would enhance clarity throughout the paper.

Answer: We decided not to use the acronyms VS, OS, and WB throughout the text. This is also in consideration of the fact that Table 2 was moved to the Supplementary Material according to the suggestion of reviewer #2.

  1. Remove parenthesis for “MPXV (and other…” in line 132, as it is unnecessary since there is no closing parenthesis for the statement.

Answer: We removed the parenthesis. See line 132 of the revised manuscript.

Reviewer 2 Report

1. In the introduction, lines 37-38, the period of 25893 mpox cases in Europe should be defined, it currently reads as possibly a daily number of cases which is misleading. 

2. In materials and methods  lines 78-81 is one long sentence. Revise into separate sentences.

3. In the results section. paragraph one, lines 143-151. The data presented is difficult to follow. revise the presentation of the data.  

4. Lines 152-155  Table 1. since the alternate diagnosis of HSV/VSV/EV/ etc. was not confirmed in testing in this study, this must be made explicit (clear) in the reporting of these results. 

5. I am unclear of the value of reporting the results collected overtime, in Table 2. as there is no significant difference between the results obtained here compared to the reference assay. This data is supplementary and can be summarized in a single statement. 

6. The value of testing different samples is also not clear. I could not find any reference to the preferred sample type for the QIAstat-DX assay and since the result on all samples tested with reference assay are in 100% agreement it does not indicate bias in any sample type.  

1. Generally sentences are long and difficult to follow. 

2. English language moderation is recommended. 

Author Response

  1. In the introduction, lines 37-38, the period of 25893 mpox cases in Europe should be defined, it currently reads as possibly a daily number of cases which is misleading.

Answer: We added the period during which the 25893 mpox cases in Europe were identified. See lines 36 to 39 of the revised manuscript.

  1. In materials and methods lines 78-81 is one long sentence. Revise into separate sentences.

Answer: We splitted the above indicated sentence in two separate sentences. See lines 78 to 81 of the revised manuscript.

  1. In the results section. paragraph one, lines 143-151. The data presented is difficult to follow. revise the presentation of the data.

Answer: We improved the paragraph above to clarify the presentation of the data. See lines 145 to 151 of the revised manuscript.

  1. Lines 152-155 Table 1. since the alternate diagnosis of HSV/VSV/EV/ etc. was not confirmed in testing in this study, this must be made explicit (clear) in the reporting of these results.

Answer: We added a sentence to state that testing to confirm the diagnosis of HSV/VSV/EV/ etc. was not performed in our study. See lines 150 to 151 of the revised manuscript.

  1. I am unclear of the value of reporting the results collected overtime, in Table 2. as there is no significant difference between the results obtained here compared to the reference assay. This data is supplementary and can be summarized in a single statement.

Answer: We moved Table 2 to the Supplementary Material (see Table S1). As a result, the former Table S1 was renamed Table S2 as well as the former Table 3 was renamed Table 2.

  1. The value of testing different samples is also not clear. I could not find any reference to the preferred sample type for the QIAstat-DX assay and since the result on all samples tested with reference assay are in 100% agreement it does not indicate bias in any sample type.

Answer: We added a sentence to address this relevant issue. See lines 285 to 289 and line 293 of the revised manuscript.